# Blood Phytosterol Concentration and Genetic Variant Associations in a Sample Population

**DOI:** 10.3390/nu16071067

**Published:** 2024-04-05

**Authors:** Leticia Garrido-Sanchez, Elisabet Leiva-Badosa, Josep Llop-Talaveron, Xavier Pintó-Sala, Toni Lozano-Andreu, Emili Corbella-Inglés, Pedro Alia-Ramos, Lluis Arias-Barquet, Josep Maria Ramon-Torrel, Maria B. Badía-Tahull

**Affiliations:** 1Pharmacy Department, Hospital Universitari Bellvitge, IDIBELL, Universitat Barcelona, 08907 L’Hospitalet de Llobregat, Spain; 2Cardiovascular Risk Unit, Internal Medicine, Hospital Universitari de Bellvitge, IDIBELL, Universitat Barcelona, 08907 L’Hospitalet de Llobregat, Spain; xpinto@bellvitgehospital.cat (X.P.-S.);; 3Pharmacy Department, Institut Català d’Oncologia, IDIBELL, Universitat Barcelona, 08907 L’Hospitalet de Llobregat, Spain; 4Clinical Laboratory Department, Hospital Universitari Bellvitge, IDIBELL, Universitat Barcelona, 08907 L’Hospitalet de Llobregat, Spain; 5Ophthalmology Department, Hospital Universitari Bellvitge, IDIBELL, Universitat Barcelona, 08907 L’Hospitalet de Llobregat, Spain; 6Preventive Medicine Department, Hospital Universitari Bellvitge, IDIBELL, Universitat Barcelona, 08907 L’Hospitalet de Llobregat, Spain

**Keywords:** phytosterols, SNVs, campesterol, sitosterol, ABCG5/8

## Abstract

The main objective of this study was to determine plasma levels of PS and to study SNVs rs41360247, rs4245791, rs4148217, and rs11887534 of *ABCG8* and the r657152 SNV at the ABO blood group locus in a sample of a population treated at our hospital, and to determine whether these SNVs are related to plasma PS concentrations. The secondary objective was to establish the variables associated with plasma PS concentrations in adults. Participants completed a dietary habit questionnaire and a blood sample was collected to obtain the following variables: campesterol, sitosterol, sitostanol, lanosterol, stigmasterol, biochemical parameters, and the SNVs. In addition, biometric and demographic variables were also recorded. In the generalized linear model, cholesterol and age were positively associated with total PS levels, while BMI was negatively related. For rs4245791, homozygous T allele individuals showed a significantly lower campesterol concentration compared with C homozygotes, and the GG alleles of rs657152 had the lowest levels of campesterol compared with the other alleles of the SNV. Conclusions: The screening of certain SNVs could help prevent the increase in plasma PS and maybe PNALD in some patients. However, further studies on the determinants of plasma phytosterol concentrations are needed.

## 1. Introduction

Sitosterolaemia or phytosterolaemia is an autosomal recessive disorder of lipid metabolism characterized by phytosterol (PS) accumulation that can lead to atherogenic and cardiovascular disease. Since the first time that phytosterolaemia was described [1], many studies have identified its origin in a mutation in the genes encoding a sterol efflux transporter, *ABCG5/8*; its function is the hepatic secretion and intestinal absorption of cholesterol and plant sterols. Mutations in the genes located on chromosome 2p21 and their genetic variation alter the function of the transporter and, therefore, dietary sterol blood concentrations, mainly vegetal sterols [2,3,4,5]. These two proteins, ABCG5 and ABCG8, form a heterodimer of the ATP-binding cassette (ABC) family and are in the canalicular membrane of hepatocytes and the apical membrane of enterocytes. ABCG5/8 is also responsible for the classic reverse cholesterol transport and transintestinal cholesterol excretion [6]. 

In 2010, a genome-wide association study [7] identified significant associations of plasma plant sterol concentrations with three single nucleotide variants (SNVs): rs4245791, rs41360247, and rs657152. rs4245791 and rs41360247 are genetic variants of the *ABCG8* hemitransporter gene. rs657152, at the blood group ABO locus, was linked to the group O allele and was associated with decreased plasma PS levels. In addition, alleles associated with increased PS levels displayed a significant association with an increased probability of coronary artery disease (CAD), and alleles associated with reduced PS levels were associated with reduced CAD risk [7].

Two other genetic variants, rs4148217 and rs11887534 were associated with blood PS concentrations in previous studies [8,9,10] (Table 1).

Phytosterolaemia is considered an extremely rare disorder; nevertheless, high PS levels may occur with the use of intravenous vegetable-based lipid emulsions (LEs) in patients on parenteral nutrition (PN). Several studies have shown, especially in newborns, that patients treated with PN have higher PS levels than healthy controls [13,14,15,16].

Parenteral nutrition-associated liver disease (PNALD), manifesting as liver function test (LFT) alterations, is a relevant complication associated with elevated plasma PS. Our group carried out a randomized double-blind clinical study with 19 patients on PN with vegetable-based LEs. We observed that plasma accumulation of PS and altered LFTs could be prevented with the exclusive administration of plant sterol-free LEs, such as Fish Oil-based LEs [17].

A substudy of the previous clinical trial (EudraCT Number: 2014-003597-17) showed that, depending on the allele pairing of both rs41360247 and rs4245791 SNVs, the LFT alteration in adult patients on short-term PN is conditional. We also concluded that further studies in larger series are necessary to determine the conditions under which the variants act and how they do so [18].

Therefore, genotyping the variants associated with plasma PS levels may allow us to personalize therapeutic strategies for patients treated with PN. In this context, we performed an observational study of a sample population attending our hospital to determine the prevalence of these SNVs and their relationship with plasma PS concentrations.

The main objective of this study was to measure plasma PS concentrations and to study SNVs rs41360247, rs4245791, rs4148217, and rs11887534 of *ABCG8* and the r657152 SNV at the ABO blood group locus in a sample population to determine whether these SNVs are related to plasma PS concentrations. The secondary objective was to establish the variables associated with plasma PS concentrations in adults.

## 2. Materials and Methods

### 2.1. The Study Population

The present work is an observational study conducted in an adult population recruited from the Departments of Ophthalmology and Preventive Medicine of Bellvitge University Hospital (L´Hospitalet de Llobregat, Barcelona, Spain). A total of 185 participants [18–81 years old; 94% Caucasian, 6% Latin] completed a dietary habit questionnaire, and blood samples were collected. Patients classified as chronic complex or those with advanced chronic disease were excluded. All volunteers provided written informed consent. The investigation project (PR 144/17) was approved by the Clinical Research Ethics Committee of the hospital on 25 May 2017.

### 2.2. Data Collection

The collected dependent variables were total PS, campesterol, sitosterol, sitostanol, lanosterol, and stigmasterol. Independent variables were the non-coding variants rs41360247 and rs4245791 of *ABCG8*, the missense variants rs11887534 and rs4148217 of *ABCG8*, and lastly the non-coding variant rs657152 of ABO, as well as the biochemical parameters: total cholesterol, triglycerides, α-tocopherol, gamma-glutamyl transferase (GGT), alkaline phosphatase (AP), alanine transferase (ALT), total bilirubin, creatinine, and urea. In addition, the following biometric and demographic variables were also recorded: sex, age, height (m), weight (Kg), and calculated body mass index (BMI) (kg/m^2^).

Dietary habits were collected with a questionnaire (Table A1) that included type of diet and eating habits (type of oil consumed, vegetarian/vegan, and the number of nuts, pastries, sausages, and butter/margarine consumed during the week) and intake of vitamins and PS-enriched supplements [19]. From the data obtained through the dietary questionnaire, the variable fat intake index (FI) was calculated, which included the frequency of saturated fats ingested (butter/margarine, sausages, and pastries) during the week, and scored from 0 (no saturated fats ingested) to 6, the maximum saturated fat intake (>3 times per week of three foods). The vegetal intake index (VII) was also a calculated variable following the same criteria, which included the vegetal ingested (nuts, PS-enriched supplements, and vegetarian diet). Finally, cholesterol-lowering treatments were also registered.

Blood samples were taken after an overnight fast, obtaining plasma values of the following data: total PS and their fractions (campesterol, beta-sitosterol, stigmasterol, sitostanol, and lanosterol); biochemical parameters (total cholesterol, triglycerides, α-tocopherol, LFT (GGT, AP, ALT, and total bilirubin)); renal function parameters (creatinine and urea); and genetic variants (rs11887534, rs4245791, rs41360247, rs4148217, and rs657152).

### 2.3. Analytical Determinations

-Phytosterol analysis

To determine plasma PS, blood samples were collected in 4 mL tubes of lithium heparin, kept cold at 2–8 °C for up to one hour, and then centrifuged at 2000× *g* for 10 min at 4 °C. Plasma was aliquoted into 5 mL plastic tubes and stored at −80 °C until processing. Measurements of different PS concentrations in the plasma were carried out using the UPLC-ACQUITY TQD measurement system, which uses liquid chromatography of high and rapid resolution (UPLC) coupled to tandem mass spectrometry as a measurement principle (MS/MS). We worked in the reverse-phase modality using a C18 UPLC column that allowed a faster and higher resolution of the chromatographic peaks. The mobile phase was composed of two solutions of ammonium acetate and 0.1% (*v*/*v*) formic acid, one in acetonitrile and the other in methanol, using a gradient elution. As a quality control, all the samples were two-fold analyzed [17].

-α-tocopherol analysis

To determine plasma α-tocopherol, blood samples were collected in 4 mL tubes of lithium heparin and kept cold at 2–8 °C for up to one hour. They were centrifuged at 2000× *g* for 10 min at 4 °C. Measurements of α-tocopherol concentrations in the plasma were carried out using the UPLC-ACQUITY TQD. The analytical variation (CVs) was between 6.1% (for a mean of 26.2 mmol/L) and 3.7% (for a mean of 63.7 mmol/L).

-DNA isolation and genotyping

Blood samples from all individuals were collected in tubes containing ethylenediaminetetraacetic acid, and genomic DNA was isolated from peripheral blood leukocytes by using the automated format Maxwell 16 (Promega, Madison, WI, USA) Blood DNA Purification Kit. All DNA samples were stored at 4 °C until polymerase chain reaction (PCR) applications were performed.

TaqMan assays (ThermoFisher) were used according to the manufacturer’s instructions to determine genotypes for variants rs11887534 (C__26135643_10), rs4148217 (C____375061_10), and rs41360247 (C__86448255_10). To determine genotypes for variants rs4245791 and rs657152, PCR-restriction fragment length polymorphism (RFLP) analyses were performed. Briefly, in the case of rs4245791, DNA was PCR amplified using primers: forward 5′-CGTCTGGTAGATAAGTTCTGGT-3′, in which the last G substitutes a T to create a restriction site for BstEII; and reverse: 5′-CTGGCCGGGATCTACTTTT-3′. For variant rs657152, primers were forward 5′-GCAGAATGGCTGAGAACACA-3′ and reverse 5′-TACATGCTGGAGCTGTTTGC-3′, and the amplified 195-fragment was cut with MseI (both restriction enzymes were purchased from New England Biolabs, Ipswich, MA, USA). DNA integrity is controlled by agarose gel electrophoresis; DNA concentration mean range: 40–100 ng/mL.

### 2.4. Statistical Analysis

We calculated that a random sample of 203 subjects could be enough to estimate, with a confidence of 95% and a precision of ±5%, considering the frequency of the minor allele of rs4245791 (this variant was the main objective in our initial study) [7]. The percentage of necessary replacements was expected to be 20%.

A descriptive statistical analysis was performed using frequency tables for all variables. For continuous variables, descriptive parameters such as *n*, mean, and standard deviation (SD) were used. For categorical variables, grouped percentages were given and a chi-square analysis was carried out. The sample was stratified based on the percentile 50 and a comparison was made between the two groups above and below the percentile 50. To evaluate the presence of an association between the categorical variables (nuts, pastries, and sausages) and PS, we incorporated the linear-by-linear association (LLA) test in the Crosstab. Simple linear regression tests were performed for continuous variables. To study the relationship between plasma concentrations of PS and fractions and variants, a one-way ANOVA analysis was performed. 

Finally, three generalized linear models were run, one for each of the following dependent variables: total PS, sitosterol, and campesterol, including the independent variables and using a linear equation minimizing residual deviance. In the case of heteroscedasticity in the residual of the model, the robust estimator was used. The overall significance of the model was established with the Omnibus test χ^2^. Models without statistical significance were refused. As complementary information, in all three models, a Bonferroni simultaneous multiple comparison test was performed.

Data were analyzed using IBM SPSS 28.0; statistical significance was reported with a 95% confidence interval (CI) at the conventional *p* < 0.05 (two-tailed).

## 3. Results

A group of 185 volunteers were recruited for the study. Plasma levels of PS were (mean ± SD): total phytosterols 1.94 ± 1.48 µcg/mL, campesterol 0.85 ± 0.91 µcg/mL, sitosterol 0.85 ± 0.73 µcg/mL, stigmasterol 0.10 ± 0.13 µcg/mL, lanosterol 0.09 ± 0.09 µcg/mL, and sitostanol 0.02 ± 0.02 µcg/mL.

Table 2 shows the demographic, biometric, dietary, and analytical data of the study individuals separated into two groups: one with PS levels above and the other below the median (P50).

The variant frequency of the population studied is shown in Table 3. All genotype frequency distributions were in Hardy–Weinberg equilibrium.

Of the five SNVs studied, only rs4245791, rs4148217, and rs657152 had data for the minor homozygous alleles. Total cholesterol, total PS, and campesterol plasma concentrations are shown in Figure 1. Differences between PS and fractions and SNVs rs41360247 and rs11887543 were not statistically significant (Appendix A).

The relation between PS, categorical, and continuous variables is depicted in Appendix A.

In the generalized linear model, cholesterol and age were positively associated with total PS, while BMI was negatively associated. Also, α-Tocopherol tended towards a negative association with campesterol (Table 4).

There were no homozygous minor allele carriers for rs41360247 and rs11887534, so these variants were excluded from the multivariant analysis. For rs4245791, T homozygotes had significantly lower campesterol concentrations compared with C homozygotes. For rs4148217, CA heterozygotes presented lower sitosterol and total PS levels versus major allele homozygotes (CC), as a trend. Finally, GG homozygotes of rs657152 had the lowest levels of campesterol compared with the other alleles of the SNV.

In the Bonferroni simultaneous multiple test, the comparison of blood PS concentrations and genetic variants displays a similar trend toward statistical significance (Appendix A).

## 4. Discussion

Genetic approach: SNVs and PS: Our study evidences a relationship between some variants in the genes encoding ABCG8 and the blood group ABO locus and PS plasma levels. This supports the hypothesis that PS levels are under tight genetic control. In previous studies from other authors, a genome-wide association showed that campesterol and sitosterol were higher in rs41360247 major allele homozygote and rs4245791 [20] and rs657152 [7] minor allele homozygotes. Silbernagel [21] described similar data, and three more SNVs (rs4299376, rs6576629, and rs4953023), not included in our study, were also studied.

In other studies, the minor allele carriers for rs11887534 showed lower plasma campesterol, sitosterol [22], LDL, and total cholesterol levels than wild-type subjects [10]. The sitosterol to cholesterol ratio for rs4148217 (T400K) minor allele carriers was lower than wild type [8]. Plat et al. found that the association between variants and PS levels was allele-dependent. Moreover, subjects with higher PS concentrations presented the homozygous wild-type rs4148217 genotype and had the greatest reductions in sitosterol levels after four weeks of plant stanol ester consumption [9] compared with carriers of minor alleles. Helgadottir´s study found an association between rs4148217, rs11887534, PS levels, and the risk of CAD. Nine variants of *ABCG5/8* were associated with PS levels [12].

In our series, as in the studies described, homozygous T carriers of rs4245791 had lower levels of PS, reaching significant differences for campesterol; for rs657152, campesterol levels in GG individuals were significantly lower than those in subjects with mutant alleles.

The SNV rs4148217 only showed a tendency toward a lower plasma sitosterol concentration in heterozygous individuals. Homozygous minor allele carriers had higher levels of campesterol than the others, the reverse of previous reports. This effect could be explained by the limited number of subjects and the presence of outlier data in the AA group.

Hepatic function and PS: Case-control, randomized cross-over intervention, or cross-sectional studies were performed to study the relation between PS and coronary heart disease (CHD) risk. Different results were reported: a significant association in some cases, no relation in others, and a protective effect of PS on CHD development was also described [23,24,25,26,27,28].

While initial studies tried to demonstrate the association between PS and CHD, an association between PS and liver function alterations has also been reported under certain conditions. In fact, high PS levels have been reported in patients receiving PN due to the administration of vegetable oil-based LEs. Ellegard et al. quantified the plasma PS levels of 21 healthy adult controls and 24 adult patients with short bowel syndrome receiving PN or not. Mean plasma PS levels were 23 µmol/L in controls, 11 µmol/L in short-bowel patients not receiving PN, and 63 µmol/L in short-bowel patients receiving PN [29]. Our group published a study of twenty-seven adult intestinal failure patients on home PN vs. seven adult controls and found PS levels of 55.4 ± 6.2 µg/mL vs. 14.8 ± 2.3 µg/mL, respectively; all LFT variables studied showed a statistical association with plasma PS in patients on PN [30]. GGT and ALT increase the association with PS in adults on PN and hepatic alterations were also seen [17]. In our data, ALT and AP did not show significant differences in the multivariate model. 

Concerning SNVs, data obtained from a substudy in PN patients found that increases in AP were associated with the T allele of rs41360247 [18]. We must consider that subjects in our sample population were not on PN and had LFTs within the normal reference range.

It is also relevant to highlight that plasma α-tocopherol tends towards an inverse association with campesterol levels; however, Gylling found a positive association between plant sterols and α-tocopherol, suggesting that higher cholesterol absorption efficiency can also raise plasmatic α-tocopherol levels [31]. On the other hand, the antioxidant effect of α-tocopherol can be useful in treating non-alcoholic steatohepatitis [32], and the addition of α-tocopherol to LEs has been applied to avoid peroxidation due to polyunsaturated fats. We included α-tocopherol in the analysis for its hepatic protection; however, subjects did not present hepatic disease, thus no conclusions can be drawn [33,34].

Demographic and nutritional factors: We found a positive association between age and total PS and campesterol, as shown in some articles. Other factors related to plasma plant sterol levels could be diet, availability of transport vehicles, and hepatic uptake [24,35]. 

The principal dietary sources of PS are vegetable-based oils, containing higher concentrations by weight than other oils, fruit, vegetables, nuts, and cereals. Many factors affect its bioavailability and bioactivity: sterol type (saturated or glycosylated), source, chemical structure, cooking, and other food ingredients [36]. Related to dietary habits, sitosterol adjusted by cholesterol has shown gradual increases with PS intake [37]. We conducted a dietary questionnaire of the volunteers to include the contribution of diet to PS levels. The ability of PS to reduce LDL levels [38,39,40] leads to functional foods with plant sterols/stanols and it could be useful in some situations [41]; therefore, for this reason, we have also included stanol-enriched food intake in the dietary questionnaire.

In patients with phytosterolaemia, sitosterol concentrations can reach >80 µg/mL [42,43]. None of the volunteers presented phytosterolaemia, the maximum sitosterol concentrations were 5.73 µg/mL in women and 2.63 µg/mL in men. A higher PS-level association with women had been previously demonstrated [43,44].

With a view to evaluate the effect of PS on the lipid profile, cardiovascular risk, and metabolic syndrome, many studies have determined that plasma campesterol and sitosterol were associated with PS intake, high absorption efficiency of cholesterol, and were inversely correlated with BMI, plasma glucose, and triglycerides [10,37]. In our study, BMI presents a negative association with PS, in addition, the group of volunteers with PS levels below the median had 1.52 kg/m^2^ higher BMI with respect PS ≤ P50 group.

Cholesterol: The low intestinal absorption rate of PS (<5%), in contrast with cholesterol absorption (50%) [45], also depends on the PS fraction. Campesterol and campestanol have a higher absorption rate than sitosterol (9.6% and 12.5%, respectively) [42]. In fact, we found that plasma total cholesterol was 500 times the value of total PS (2.01 vs. 103.1 mg/dL). Campesterol and sitosterol were the main PS fractions, while sitostanol was the minor fraction and its concentration was stable among variants.

Plasma campesterol and sitosterol levels had a positive correlation with cholesterol absorption, dietary plant sterols, and biliary cholesterol secretion, and are inversely related to cholesterol synthesis [46]. In our study, a positive association between total PS and cholesterol plasma levels, which can be explained through the absorption mechanism shared by both compounds, was found. In addition, the difference between the mean cholesterol in the groups with PS levels above and below the median (P50) was 27.6 mg/dL.

Limitations: This work is our first approach towards a target of adult patients to determine risk factors that can be considered in the clinical field, so the study has some limitations that should be considered for future studies. The limited number of subjects did not allow us to obtain data from homozygous minor allele individuals in two of the five SNVs studied (rs41360247 and rs11887534), and for the other variants, the number of minor allele homozygotes was insufficient to obtain strong associations.

The analytical methods and biological variation are essential, and become important for comparing studies [44].

We collected race as a variable, but 94% of the subjects were Caucasian and the rest Latin; for this reason, we decided to exclude race in the statistical analysis.

With the aim of determining dietary habits, volunteers filled out a survey with no follow-up, thus without intervention on the diet we cannot conclude that the statistically significant difference may be due to the consumption of a food or its absence. It is also relevant to highlight that we did not register the amounts of plant sterols, fat intake, and cholesterol-lowering therapy doses. For this reason, the results obtained can only be considered as trends.

## 5. Conclusions

Cholesterol, age, and BMI were associated with blood levels of PS. TT individuals for rs4245791 in *ABCG8* and GG individuals for rs657152 in ABO genes display lower blood campesterol levels. For rs4148217 in the ABCG8 gene, AC individuals tended towards lower sitosterol and total PS levels. The screening of certain SNVs could help prevent the increase in plasma PS and maybe PNALD in candidates for long-term PN programs without gastrointestinal stimuli. Further studies on the determinants of plasma phytosterol concentrations are needed.

## Figures and Tables

**Figure 1 nutrients-16-01067-f001:**
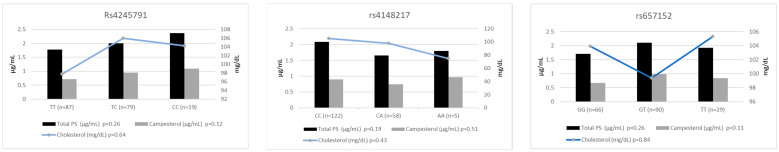
Total cholesterol, total phytosterols, and campesterol plasma concentrations for rs4245791, rs4148217, and rs657152.

**Table 1 nutrients-16-01067-t001:** Genetic control: variants, alleles, and effects.

Variants (Gene)	Ref Allele > Alt allele (Allele Frequency) [11] *	Effect of Alt Allele on Plasma PS Levels	References
rs41360247 (*ABCG8*)	T > C (C = 0.0604)	Lower levels	[7]
rs4245791 (*ABCG8*)	C > T (C = 0.3195)	Higher levels	[7]
rs4148217 (*ABCG8*)	C > A, T (A = 0.1901)	Lower levels	[8,9,12]
rs11887534 (*ABCG8*)	G > C (C = 0.0651)	Lower levels	[8,10,12]
rs657152 (*ABO*)	A > C, T (A = 0.3915)	Higher levels	[7]

* Population: European (non-Finnish); Ref: reference; Alt: alternative; PS: phytosterols.

**Table 2 nutrients-16-01067-t002:** Demographic, biometric, dietary, and analytical data of the studied subjects separated by median PS.

Variable	Total (*n* = 185) Mean ± SD/N (%)	PS ≤ P50 (*n* = 93) Mean ± SD/N (%)	PS > P50 (*n* = 92) Mean ± SD/N (%)	*p* *
**Demographic and anthropometric data**				
Age (years)	41.49 ± 18.57	42.82 ± 19.07	40.14 ± 18.07	0.164
BMI (kg/m^2^)	24.47 ± 4.28	25.23 ± 4.83	23.71 ± 3.51	**0.008**
Sex (women)	115 (62.20%)	56 (60.20%)	59 (64.10%)	0.583
**Dietary habits**				
Vegetarian-vegan (yes)	9 (4.90%)	4 (4.30%)	5 (5.40%)	0.720
Nuts				0.125 **
No consumption	58 (31.40%)	32 (34.4%)	26 (28.30%)	
1–3 times a week	99 (53.50%)	51 (54.80%)	48 (52.20%)	
>3 times a week	28 (15.10%)	10 (10.80%)	18 (19.60%)	
Pastries				0.069 **
No consumption	94 (50.80%)	43 (46.20%)	51 (55.40%)	
1–3 times a week	78 (42.20%)	40 (43.00%)	38 (41.30%)	
>3 times a week	13 (7.00%)	10 (10.80%)	3 (3.30%)	
Sausages				0.632 **
No consumption	24 (13.00%)	13 (14.00%)	11 (12.00%)	
1–3 times a week	103 (55.70%)	52 (55.90%)	51 (55.40%)	
>3 times a week	58 (31.40%)	28 (30.10%)	30 (32.60%)	
FI (0–6)	2.05 ± 1.16	2.06 ± 1.24	2.04 ± 1.07	0.451
VII (0–6)	1.22 ± 1.33	0.98 ± 1.11	1.34 ± 1.46	**0.031**
**Cholesterol treatment and dietary supplementation**				
Cholesterol-lowering treatment (Yes)	16 (8.6%)	9 (9.7%)	7 (7.6%)	0.617
PS-enriched supplements (Yes)	8 (4.30%)	2 (2.2%)	6 (6.5%)	0.144
Vitamins (Yes)	12 (6.50%)	6 (6.50%)	6 (6.50%)	0.985
**Plasma values**				
Total cholesterol (mg/dL)	101.92 ± 57.65	88.18 ± 54.63	115.81 ± 57.57	**<0.001**
Alanine aminotransferase (µKat/L)	0.29 ± 0.13	0.29 ± 0.14	0.28 ± 0.12	0.323
Alkaline phosphatase (µKat/L)	0.98 ± 0.28	1.00 ± 0.30	0.96 ± 0.25	0.184
α-Tocopherol (µmol/L)	30.25 ± 6.20	30.12 ± 5.99	30.39 ± 6.42	0.380
Creatinine (µmol/L)	76.82 ± 16.83	78.24 ± 17.67	75.38 ± 15.89	0.125
Albumin (g/L)	47.74 ± 3.29	47.82 ± 3.34	47.66 ± 3.26	0.375

FI: fat intake index; VII: vegetal intake index; PS > PS50: phytosterol concentrations above the median; PS ≤ P50: phytosterol concentrations less than or equal to the median; * differences between PS ≤ P50 and PS > P50; ** statistical significance for the linear trend measure in the PS > P50 group vs. PS ≤ P50 group.

**Table 3 nutrients-16-01067-t003:** Single nucleotide variant frequency of the *ABCG8* gene and the blood group ABO locus in the study population.

Polymorphism	Genotypes (N/%)		
rs41360247	CC	TC	TT
	0 (0%)	26 (14.1%)	159 (85.9%)
rs4245791	CC	TC	TT
	19 (10.3%)	79 (42.7%)	87 (47.0%)
rs4148217	AA	CA	CC
	5 (2.7%)	58 (31.4%)	122 (65.9%)
rs11887534	CC	GC	GG
	0 (0%)	27 (14.6%)	158 (85.4%)
rs657152	TT	GT	GG
	29 (15.7%)	90 (48.6%)	66 (35.7%)

**Table 4 nutrients-16-01067-t004:** General linear model: PS and variables.

Variable	Total PS		Sitosterol		Campesterol	
	B [95% CI]	*p*	B [95% CI]	*p*	B [95% CI]	*p*
Cholesterol (mg/dL) (×10^4^)	10.00 [0.00–10.00]	**<0.001**	0.00 [0.00–10.00]	**<0.001**	0.00 [−0.18–10]	0.065
rs4245791_TT vs. rs4245791_CC	−0.57 [−1.28 –0.15]	0.119	−0.16 [−0.47–0.15]	0.311	−0.48 [−0.93–(−0.03)]	**0.037**
rs4245791_TC vs. rs4245791_CC	−0.41 [−1.10–0.28]	0.244	−0.23 [−0.52–0.07]	0.137	−0.20 [−0.63–0.22]	0.344
rs4148217_AA vs. rs4148217_CC	0.10 [−1.16–1.36]	0.880	−0.16 [−0.70–0.39]	0.577	0.40 [−0.33–1.13]	0.282
rs4148217_AC vs. rs4148217_CC	−0.38 [−0.83–0.06]	0.093	−0.19 [−0.38–0.01]	0.059	−0.12 [−0.42–0.18]	0.443
rs6577152_TT vs. rs6577152_GG	0.29 [−0.34–0.92]	0.366	0.08 [−0.20–0.35]	0.584	0.26 [0.02–0.51]	**0.035**
rs6577152_GT vs. rs6577152_GG	0.52 [0.07–0.96]	**0.022**	0.17 [−0.02–0.36]	0.074	0.36 [0.07–0.65]	**0.016**
Sex (women)	0.20 [−0.34–0.74]	0.477	0.19 [−0.04–0.42]	0.109	−0.02 [−0.43–0.38]	0.907
Age (years) (×10)	0.23 [0.09–0.37]	**0.002**	0.02 [−0.04–0.08]	0.491	0.17 [0.09–0.25]	**<0.001**
BMI (kg/m^2^) (×10)	−0.57 [−1.10–(−0.04)]	**0.034**	−0.25 [−0.47–(−0.02)]	**0.033**	−0.37 [−0.69–(−0.04)]	**0.029**
Cholesterol-lowering treatment (Yes)	−0.10 [−0.94–0.74]	0.813	0.05 [−0.31–0.41]	0.791	0.04 [−0.43–0.51]	0.870
Vitamins (Yes)	−0.17 [−1.00–0.65]	0.685	0.04 [−0.32–0.40]	0.827	−0.19 [−0.58–0.20]	0.349
Alanine aminotransferase (µKat/L)	0.22 [−1.45–1.90]	0.794	0.27 [−0.46–1.00]	0.468	−0.20 [−0.89–0.49]	0.566
Alkaline phosphatase (µKat/L)	0.58 [−0.22–1.38]	0.152	0.12 [−0.23–0.46]	0.512	0.50 [−0.34–1.33]	0.247
Creatinine (µmol/L) (×10^4^)	−80.00 [−220.00–70.00]	0.303	0.01 [−60.00–60.00]	1.000	−80.00 [−180.00–20.00]	0.127
FI	−0.06 [−0.24–0.12]	0.494	−0.04 [−0.12–0.04]	0.300	−0.04 [−0.16–0.09]	0.551
VII	0.03 [−0.13–0.18]	0.755	0.03 [−0.03–0.10]	0.334	−0.02 [−0.10–0.07]	0.723
α-Tocopherol (µmol/L) (×10)	−0.21 [−0.55–0.13]	0.222	−0.02 [−0.17–0.12]	0.757	−0.19 [−0.39–0.01]	0.057
Omnibus χ^2^	41.38	**<0.001**	54.50	**<0.001**	43.95	**<0.001**
R^2^	0.20		0.21		0.25	

FI: fat intake index; VII: vegetal intake index.

## Data Availability

Data is contained within the article (and Appendix A).

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
