# Peer review of "Blood Phytosterol Concentration and Genetic Variant Associations in a Sample Population"

_nutrients, 2024, doi:10.3390/nu16071067_

Round 1

Reviewer 1 Report

Comments and Suggestions for Authors

The authors investigated how clinical, demographical, and genetic variants in ABCG5/8 genes associated with phytosterols (PS) levels in blood in a sample population from their reference area. There are several major concerns, which the authors need to address.

1. The race and ethnicity about the study population is unclear and not considered in the association analysis, which is important in genetic association study.

2. The statistical method used in the analysis can be improved, such as the Cochran-Armitage trend test and additive genetic model.

3. The multiple testing problem is not considered in statistical analysis.

4. Lack of replication study of their findings.

5. The two paragraphs at the end of Method section are not relevant to the manuscript.

Comments on the Quality of English Language

NA

Reviewer 2 Report

Comments and Suggestions for Authors

A manuscript entitled “Blood Phytosterol Concentration and Genetic Variant Associa- 2 tions in a Sample Population” by Garrido-Sanchez et al. is very interesting and provides a valuable data.

However, I have a few major and minor concerns:

-          Abstract: In my opinion, it is not written clearly enough. Please state first what the aim of the study was, which variables you tested, and then the most important results.

-          Line 27: It is not clear to me what “reference area” means?

-          Line 52: Please provide the reference.

-          Line 65: Please edit references [13-16]

-          Line 89: Please provide more information about the participants like number of participants, age range etc. Were the data collected just in one hospital? If so, please provide the name of the city and state.

-          Line 93: Please provide the relevant information regarding ethical approval.

-          Statistical analysis:

o   Did you check for the normality of the continuous variables? If so, which test did you use? Please provide results. This is crucial for the choice of statistical tests.

o   Did you calculate the power of the study for given sample size?

-          Please provide data for LD between analyzed SNPs (r2, D).

-          Please add information about the type of SNVs you analyzed (e.g. missense variant, non-coding variant, etc.).

-          Line 163-Line 179: Delete text.

-          Table 2: I think it would be better to extract the variables Total Phytosterols- Sitostanol (the ones that were not compared between 2 groups) from the table and describe them in the main text.

-          Line 194: Add Hardy-Weinberg equilibrium.

-          Line 331-335: Delete text.

Comments on the Quality of English Language

Minor editing of English language required

Reviewer 3 Report

Comments and Suggestions for Authors

The authors describe the association between blood phytosterol concentrations and genetic variants and demographic, biometric, dietary, and analytical data of the studied, overall healthy, subjects. They applied two types of analysis. One is comparison of potentially affecting parameters in two groups of plasma phytosterol concentrations, i.e. below P50 and above P50. The second to calculate linear correlations. The authors provide interesting data.

However, the interpretation and explanation of data are incomplete or lacking. Supplemental data are not discussed in the manuscript. However, the supplemental data supply the same parameters, but now  calculated based on reversed statistics compared with the statistics in the manuscript and led to other conclusions. The authors conclusions are the combination of both approaches. This is very confusing. The authors should clarify their principle of using two sets of data obtained with different statistics and explain the logic how to draw conclusions or they should stick with one statistical approach.

Comments:

Methods section describes phytosterol analysis, α-tocopherol analysis and DNA isolation and Genotyping. No quality control data are described and no references are supplied referring to the original method description and evaluation. This information must be supplied.

Line 164 to 179 describe the journals requirements for the materials and methods section. This must be removed. The same observation counts for lines 331 to 335.

A group of 185 volunteers was included in the study. In the discussion, the authors describe the limitations of the study and mention the small sample size. Indeed, the group size is small when describing so many different parameters. To overcome the problem, the authors defined two experimental subgroups having a serum PS concentration < or > P50.   All the analyzed parameters are considered independent parameters. This may not be true. Specific imaginary subgroups like overweight oldest women eating many sausages may have extreme high or low PS values. This detailed information cannot be obtained in the small group of subjects.

Table 2 indicates only three parameters that affect the PS level. The first is BMI, but interestingly, the difference between the PS levels <P50 and > P50 relates to a difference in BMI of only 1.5 kg/m2 or about 6%. Furthermore, the PS and BMI are negatively associated (table 4). A second biochemical parameter is total cholesterol. Here the difference is 27 mg/dl or about 27% when related to the mean of the total group. This association is positive and may be linked to the shared mechanism of absorption for cholesterol and PS.

Line 317 states that a positive association between PS and cholesterol was found. However, table 4 indicates that this association holds only for sitosterol and not for campesterol. Also, the associations between rs6577152_TT vs. rs6577152_GG and rs6577152_GT vs. rs6577152_GG and PS indicate differences between sitosterol and campesterol. Can the authors explain the different effects of the genetic differences for both phytosterols? The genetic difference should affect the transporting proteins that affect both phytosterols.

In table 2, age appears to have no effect on PS concentrations. However, in table 4, age and PS show a significant linear correlation, i.e. for total PS and campesterol, not for sitosterol. This point needs discussion. How do the authors explain the difference between group comparison (table 2) and linear association (table 4) and the difference between sitosterol and campesterol?

The authors studied the effect of cholesterol lowering therapy and found no effects on plasma PS. Did therapy only include statin treatment? If so, they should state statin treatment. Ezetimibe treatment should lead to altered plasma PS levels.

The authors conclude that dietary habits affect plasma PS concentrations. This is not indicated by P-values in table 2. Even intake of PS enriched supplements did not cause an increase. They also state that gender is a determinant of plasma PS. This is not true according to the data in tables 2 and 4.

Otherwise, the conclusions are supported by the supplemental data. Strangely, here, the authors present the same parameters, but calculated with a reversed statistics. For example, PS values are compared between men and women. This way a significant difference is observed. This is in contrast to the results obtained with the approach described in the manuscript, where the PS values were separated in a <P50 and >P50 group and gender was compared in the two groups. How to conclude when from two approaches one leads to a significant difference. To this reviewer’s opinion, the approach used in the supplemental data is a more accepted approach than the one published in the manuscript. It is my advice to apply the supplemental data into the manuscript and to build the discussion around these data. 

Round 2

Reviewer 1 Report

Comments and Suggestions for Authors

Thanks for addressing the previous comments.

1. The Cochran-Armitage trend test can be applied in Table 2, such as Pastries vs PS group.

2. In Table 4, which dependent variable was analyzed using generalized linear model? Could you please specify the outcome level, if categorial or ordinal variables were analyzed?

3. The authors' response on multiple-testing is not acceptable. Shown in Table 4, there are 17 independent variables included in the regression model. The family-wise type I error is inflated without multiple-testing correction. It is true that type II error will increase when we consider multiple-testing. But family-wise type I error of 0.05 is considered a good trade-off between type I error and type II error. If the authors think power is more important, please discuss the inflated type I error with real numbers. Further, there is no replication to support the power benefitting from the uncontrolled family-wise type I error.

Reviewer 2 Report

Comments and Suggestions for Authors

I have no further comments.

Comments on the Quality of English Language

 Minor editing of English language required

Author Response

Thank you very much again for taking the time to review this manuscript. 

We have reviewed the document and made some minor language corrections.

[Line 81] [Line 182]

Reviewer 3 Report

Comments and Suggestions for Authors

By focussing on the multivariate model the conclusions now became straightforward. Therewith, the manuscript has clearly improved.

Otherwise, the authors mean to introduce reference 20 to validate the methodology used for sterol analysis. However, the authors use a modern UPLC-ACQUITY TQD technology, whereas reference 20 describes a GC/MS procedure as was common in1993. This is annoying. Did the authors develop and describe the methodology in the literature, please refer. Did they copy the procedure from the literature, please refer. If not, please describe the quality control data. 

Round 3

Reviewer 1 Report

Comments and Suggestions for Authors

Thanks for all the revisions. Could you please incorporate Response 3 into the manuscript, especially the last table in Response 3.
